# Comparison of two proxies for the preconception weight using data from a pre-pregnancy cohort in Benin: Weight measured in the first trimester of pregnancy vs estimated by Thomas' formula

Emmanuel Yovo[1,2]*, Manfred Accrombessi[1,3], Valérie Briand[4,5], Gino Agbota[1,6], Cornelia Hounkonnou[1], Jules Alao[7], Jennifer Zeitlin[8], Pierre Traissac[2], Yves Martin-Prevel[2]

1 Institut de Recherche Clinique du Bénin (IRCB), Abomey-Calavi, Benin, 2 Institut Agro, UMR MoISA, University Montpellier, CIRAD, CIHEAM-IAMM, INRAe, IRD, Montpellier, France, 3 Malaria International Department, Population Services International (PSI), Cotonou, Benin, 4 UMR MERIT, University Paris Cité, IRD, Paris, France, 5 Epicentre, Paris, France, 6 UMR TransVIHMI, University Montpellier, IRD, INSERM, Montpellier, France, 7 Service de Pédiatrie, Centre Hospitalier Universitaire de la Mère et de l'Enfant-Lagune de Cotonou, Cotonou, Benin, 8 Centre of Research in Epidemiology and Statistic (CRESS), University Paris Cité, University Sorbonne Paris Nord, Inserm, INRAe, Paris, France

* emkoffiyovo@gmail.com

## Abstract

Accurate determination of pre-pregnancy weight is essential for optimal pregnancy monitoring and antenatal care. Determining pre-pregnancy weight in limited-resources settings is challenging for both clinical practice and public health research. From a 2014–2017 pre-pregnancy cohort in Benin, we evaluated the agreement between the measured pre-pregnancy weight (MPPW) and two proxies: (i) the first trimester pregnancy weight (FTPW) and (ii) the estimated pre-pregnancy weight (EPPW) using Thomas & al. formula. We analysed data from 302 pregnant women with both pre-pregnancy weight measured within 3 months before conception and weight measured during the first trimester. Using segmented linear regression, we first assessed up to which gestational age the weight measured during the first trimester could reasonably estimate the MPPW. Then the Bland & Altman method was used to assess agreement between MPPW and the two proxies. Additional analyses were performed to assess the sensitivity of results to the timing of measurement of either MPPW or the two proxies. On average, FTPW did not feature significant difference with MPPW up to 13.03 (11.99–14.06) weeks of gestational age. FTPW, measured on average at 7 ± 2.4 weeks of gestation, and the EPPW showed similar Bland & Altman limits of agreement with the MPPW. However, while the FTPW slightly underestimated the MPPW by a mean of—0.16 (-0.08; +0.39) kg, the EPPW overestimated it by a mean of + 0.43 (+0.20; +0.66) kg. Minor differences in these results were observed when the MPPW was assessed earlier or within three months before pregnancy, or according to the gestational age at the time of the proxy's measurement. In conclusion, in Southern Benin and up to 12–14 weeks of

**Data Availability Statement:** All relevant data are within the manuscript and its Supporting information files.

**Funding:** This work was supported by the French Agence Nationale de la Recherche (ANR-13-JSV1-0004) grant to Dr Valérie Briand and the Foundation Simone Beer under the auspices of the Fondation de France (grant number 00074147) to Dr Valérie Briand. Dr Emmanuel Yovo received the grant ARTS (Allocations de Recherche pour une Thèse au Sud, N/Réf.: PDG-MAPS- 2020-308), the original French name of a PhD grant program from the French National Research Institute for Sustainable Development (IRD) for a PhD study at Montpellier University.

**Competing interests:** All authors report no potential conflict of interest.

pregnancy, the FTPW appeared to be a good proxy of the MPPW while using Thomas' formula did not enhance pre-pregnancy weight estimation.

## Introduction

A key factor influencing gestational weight gain and its impact on maternal and foetal health is the woman's weight at the beginning of pregnancy [1,2]. In 2009, the Institute of Medicine (IOM) revised its guidelines to establish optimal gestational weight gain based on the woman's pre-pregnancy body mass index (BMI) category [3]. Several studies demonstrated that a weight gain below or above the recommended ranges was associated with adverse pregnancy outcomes such as increased rates of caesarean section, interventional delivery or need for neonatal intensive care [4–6]. At the individual level, accurate knowledge of a woman's pre-pregnancy weight is essential for optimizing pregnancy monitoring and clinical decision-making [7]. At the population level, for research purposes or to inform public health programs, an accurate estimate of the pre-pregnancy weight enhances the reliability of studies that evaluate the effectiveness of interventions or for surveillance purposes [8,9]. However, the availability of accurate pre-pregnancy weight is challenging in both routine antenatal care and in population studies. Also, as an individual's weight fluctuates over time, there is the question of how long before pregnancy a weight can be considered as a reference pre-pregnancy weight. To our knowledge, this point is rarely examined, while it may impact the results of any study.

There are three commonly used methods, in research studies as well as in clinical practice, to estimate the pre-pregnancy weight when it is unknown: (i) Relying on self-reported pre-pregnancy weight provided by pregnant women themselves. However, research has abundantly shown that women tend to underestimate their pre-pregnancy weight when reporting it from memory, particularly when they are overweight or obese [10–13]. (ii) Using the weight measured during the first trimester of pregnancy as a proxy for pre-pregnancy weight. While this approximation is understandable, its validity has been assessed in only a few studies, mainly conducted in high income countries [14,15]. (iii) Recently, Thomas and colleagues proposed a formula to estimate pre-pregnancy weight, also using the weight measured during the first trimester of pregnancy, but accounting for the following parameters: gestational age at the time of weight measurement, height, parity and age of the woman at the time of pregnancy [16]. To our knowledge, only one study carried out in UK compared the results of these methods against a measured pre-pregnancy weight [17]. It showed that both weight measured in early pregnancy and weight estimated using Thomas' method were on average 0.88 kg higher than the actual pre-pregnancy weight, with a fair agreement, while the self-recalled pre-pregnancy weight underestimated the actual weight and exhibited a lower agreement.

Most of the studies exploring methods for estimating pre-pregnancy weight were performed in high- or upper-middle-income countries. This raises the question of their validity in low-resources settings, particularly in Sub-Saharan Africa (SSA) where, to the best of our knowledge, no study has addressed this issue while self-reported pre-pregnancy weight is likely to be very inaccurate in this type of context, due to the low socioeconomic and educational levels of many women of childbearing age. In addition, retrieving women's weight from medical charts is not an option since women are rarely weighed during medical consultations before pregnancy. Furthermore, pregnant women generally attend the maternity clinics late [18,19], while factors associated with inappropriate weight gain during pregnancy are highly prevalent in SSA [20–22].

This study intends to fill this lack of knowledge through a secondary analysis of data from a pre-conceptional cohort in Benin. We aimed to assess the agreement between the measured pre-pregnancy weight (MPPW) considered as the reference weight, the weight measured in the first trimester of pregnancy (FTPW) and the estimated pre-pregnancy weight using Thomas formula (EPPW).

## Materials and method

### Study settings

Benin is a West African country with a population of approximately 12 million people, ranked 166[th] out of 191 countries according to Human Development Index (HDI) of 2021–2022 [23]. RECIPAL, the pre-pregnancy cohort study, recruited participants from two districts in southern Benin: Akassato and So-Ava. So-Ava is nestled in a riverine environment and relies heavily on fishing and agriculture in a rural context. Akassato is located on the mainland in a semi-urban setting and is home to many people working in Cotonou and Abomey-Calavi, the two largest towns nearby.

### Procedures

The RECIPAL pre-conceptional cohort was carried out from 2014 to 2017 and was primarily designed to study the consequences of malaria infection in early pregnancy. Its detailed protocol has already been described elsewhere [24]. Briefly, women of reproductive age (18–45 years old) were recruited at the community level, then followed monthly for a maximum period of 24 months until becoming pregnant. At inclusion, the study participant's anthropometric, demographic, clinical and socioeconomic characteristics were collected. The pre-pregnancy follow-up consisted of monthly home visits at which the first day of the last menstrual period (LMP) was recorded and a urinary pregnancy test was performed. The women's weight was measured every three months during the first year of follow-up. The subsample of women who became pregnant was then followed monthly from early pregnancy to delivery.

The RECIPAL study received ethical approval from the president of the Beninese Ethics Committee of the Institut des Sciences Biomedicales Appliquées and Ministry of Health (decision no. 39 of 05/16/ 2014). The community-based recruitment of adult women desiring pregnancy began on June 11, 2014, and ended on April 4, 2016. The follow-up of women who became pregnant concluded on August 31, 2017. All participants gave informed written consent before enrolment in the cohort.

### Measurements

Anthropometric measurements were collected using standard procedures [25]. The height was measured to the nearest millimeter with a SECA 206 (Hamburg, Germany) gauge at the health facility level; the body weight was measured with a 200 g precision with calibrated electronic scales (Tefal, France) during household's visits before pregnancy and at the health facility thereafter. Ultrasound for dating the pregnancy was performed between 9 and 13 weeks of gestation (wg) (±1week); the pregnancy dating was based on the crown-rump length (CRL) measurement using Robinson's formula [26]. Gestational age (GA) was based on the LMP if the difference between the LMP and CRL was less than 7 days or on CRL otherwise. The gestational weight gains (GWG) were calculated by subtracting the measured pre-pregnancy weight from each weight measurement during pregnancy.

## Measured pre-pregnancy weight and its proxies

1. **The measured pre-pregnancy weight (MPPW)** was measured within 3 months prior to the start of pregnancy in the sample selected for the main analysis. The timing of the measurement was assessed with respect to the estimated date of conception.

2. **The First Trimester of Pregnancy Weight (FTPW)** was measured early in the first trimester of pregnancy.

3. **The Estimated Pre-pregnancy Weight using Thomas et al. formula (EPPW)** [16] was based on a measurement of weight during the first trimester of pregnancy, accounting for the gestational age at which the measurement was taken, height, parity and age of the pregnant woman as follows:

Estimated pre-pregnancy weight (kg) $= 6.10 + 0.99$ (first trimester measured weight in kg) $-$ 0.01 (gestational age (day) at first weight measurement) $- 0.02$ (height in cm) $- 0.04$ (maternal age in years) $- 0.09$ (parity)

## Statistical analyses

Data management and statistical analyses were performed using Stata 17.0 (Stata Corp). First type error rate was set at 0.05 for all analyses.

1. We estimated the upper limit of gestational age at which the weight measured during pregnancy could be used to approximate the pre-pregnancy weight:—first the trend of GWG from early pregnancy to delivery was graphed as a function of gestational age using a cubic spline model,—then, we modelled the weight gain using a 3 part segmented linear regression: both the two separation points and the 3 slopes were estimated from the data, resolving in a non-linear model. Variance estimates were adjusted for repeated measurements. The main parameters of interest where the gestational age value of the first separation point and the value of the first slope (the null hypothesis of a zero slope being that of a non-significant weight gain for a gestational age less than the first separation point).

2. We compared the proxies of the pre-pregnancy weight (FTPW and EPPW) with the measured pre-pregnancy weight (MPPW):—we used the Bland and Altman agreement method [27] and, accordingly, the results are presented as plots of the differences between the two measurements against their mean, the limits of agreement and 95% confidence intervals (95% CIs) around those limits,—we calculated the Lin concordance correlation coefficients [28]. Also, we used kappa coefficients [29] to compare classifications of women according to pre-pregnancy BMI WHO categories as estimated for the actual pre-pregnancy weight and its two proxies (FTPW and ETPPW, respectively).

3. Then (as a sensitivity analysis), we estimated the influence on the agreement between the actual pre-pregnancy weight and its two estimates of:

   *(a) the gestational age at the first trimester weight measurement.* We carried out the Bland & Altman agreement analysis on six subgroups of women: $\leq 5$ weeks, $> 5$ and $< 7$ weeks, $\leq 7$ and $< 9$ weeks, $\leq 9$ and $< 11$ weeks, $\leq 11$ and $< 13$ weeks and $\leq 13$ and $< 14$ weeks. In addition, on the women x measurement methods x visit data points, we fitted a linear model with weight as the response variable and measurement method (MPPW, FTPW, ETTPW) and FTPW time of measurement as covariates, that included also the method x time

interaction term (to estimate whether mean difference between the 3 methods would differ according to time of measurement). Variance estimates were adjusted for repeated measurements.

*(b) the time between the measured pre-pregnancy weight and the estimated date of conception.* We carried out the Bland & Altman analysis on two sub-groups of women, according to whether the MPPW was measured within three months or more than three months before pregnancy. Similarly, on the women x measurement methods data points, we also fitted a linear model with method and time of MPPW measurement as covariates and we assessed the method x time of measurement interaction (to estimate whether mean difference between the 3 methods would differ according to whether MPPW was measured before or after three months before pregnancy). Variance estimates were adjusted for repeated measurements.

## Results

As summarized in Fig 1, a total of 1,214 women of childbearing age meeting the inclusion criteria were recruited at the community level into the RECIPAL cohort for pre-pregnancy follow-up, of which 411 (33.9%) became pregnant. Of these 411 women, 302 (73%) had a pre-pregnancy weight measured within three months before getting pregnant and at least one weight measured at antenatal care visit (ANC) in the first trimester of pregnancy. The first visit during pregnancy occurred at a mean gestational age of 7±2.4 wg. Overall, women had between 1 and 3 weight measurements during first trimester of pregnancy, resulting in a total of n = 742 women x weight measurement data points which were used for sensitivity analyses.

Sociodemographic and anthropometric characteristics of the women are described in Table 1. They were 27-year-old on average and two out of three were married and monogamous. More than two thirds of the women were from Toffin ethnic group, seven out of ten were illiterate and almost all had a professional activity. The mean parity was 2.8 with 11% of nulliparous and 19% of primiparous women (Table 1). Regarding the women's

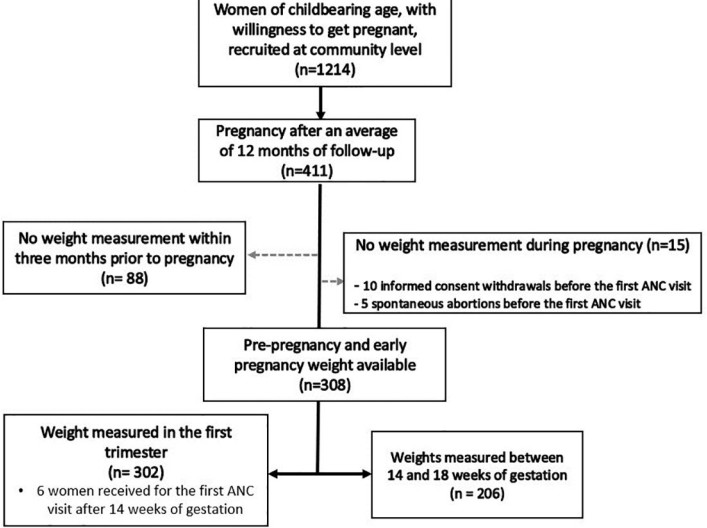

**Fig 1. Flowchart diagram of the study.** RECIPAL cohort, Southern Benin, 2014–2017.

**Table 1. Sociodemographic and anthropometric characteristics of women at inclusion in RECIPAL study in Benin, 2014–2017 (n = 302).**

| Characteristics | Unit or category | Mean ± SD or % |
|---|---|---|
| Maternal age | Year | 26.7 ± 5.1 |
| Area of residence | Sô-ava | 69.8% |
| | Akassato | 30.2% |
| Ethnic group | Toffin | 69.2% |
| | Aïzo | 16.2% |
| | Others | 14.6% |
| Marital status | Unmarried cohabitation | 5.2% |
| | Married monogamous | 65.5% |
| | Married polygamist | 29.3% |
| Education level | Illiterate | 70.1% |
| | Primary/literate | 18.8% |
| | Middle or high school or higher education | 11.1% |
| Women's professional status | Active | 93.5% |
| | Unemployed | 4.9% |
| | In training | 1.6% |
| Parity | Number | 2.8±2.02 |
| | 0 | 10.7% |
| | 1 | 18.8% |
| | $2 \leq$ and $< 5$ | 50.1% |
| | $\geq 5$ | 20.4% |
| Maternal height | cm | 157.9 ± 6.07 |
| Measured pre-pregnancy weight (MPPW) | kg | 57.2 ±11.3 |
| Pre-pregnancy BMI | kg/m² | 23.0 ± 4.30 |
| | $< 18.5$ | 9.1% |
| | $\geq 18.5$ and $< 25$ | 66.2% |
| | $\geq 25$ | 24.7% |
| Estimated gestational age at the first antenatal care visit | Weeks | 7.0 ±2.4 |
| First trimester of pregnancy weight (FTPW) | kg | 57.1 ±11.6 |
| Estimated Pre-pregnancy weight using Thomas's formula (EPPW) | kg | 57.7 ±11.3 |

anthropometric status, the mean pre-pregnancy BMI was 23.0 ±4.3 kg/m² with two out of three women having a BMI in the normal range according to WHO cut-offs and a quarter being overweight. The mean MPPW, FTPW and EPPW were 57.2, 57.1 and 57.7 kg, respectively (Table 1).

The sociodemographic and nutritional characteristics at inclusion of the 302 selected women compared to the 109 excluded ones are presented in S1 Table. The excluded women tended to be taller and larger in body size. They were more often from So-Ava than from Akassato and were more frequently of the Toffin ethnic group.

Modelling the cumulative weight gain throughout pregnancy, three different trends were identified from the graphed cubic splines, then estimated using piecewise regression analysis (Fig 2). The slope of the first segment agreed with the null hypothesis (β = -0.03, p = 0.35) corresponding to the first segment of pregnancy until 13±1 weeks, thus justifying the use of the weight measured at any point during the first trimester as a proxy of the MPPW. The other

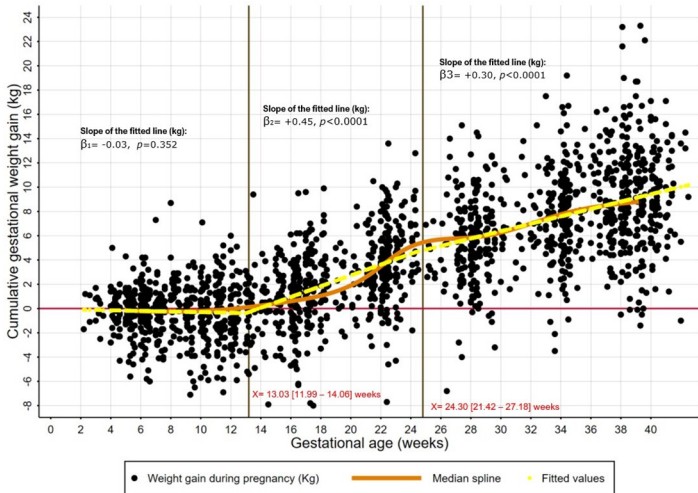

**Fig 2. Gestational weight gain trends from early pregnancy to delivery, cubic spline and piecewise regression modelling.** RECIPAL cohort, Southern Benin, 2014–2017, n = 2305 women x visit.

two slopes were positive, suggesting weight gain, first from 13 to 24 weeks at a high rate (+450 g per week), then from 24 weeks to delivery, at +300 g per week on average (Fig 2).

The results of the Bland and Altman agreement analysis are presented in Figs 3 and 4 and Table 2. The two Bland & Altman graphs were similar and showed an overall satisfactory agreement between the proxies and MPPW (Figs 3 and 4, for the FTPW and the EPPW, respectively). There were 6.62% and 5.96% of the FTPW and EPPW values that were outside the limits of agreement of the Bland & Altman graph, respectively (Table 2). The MPPW was

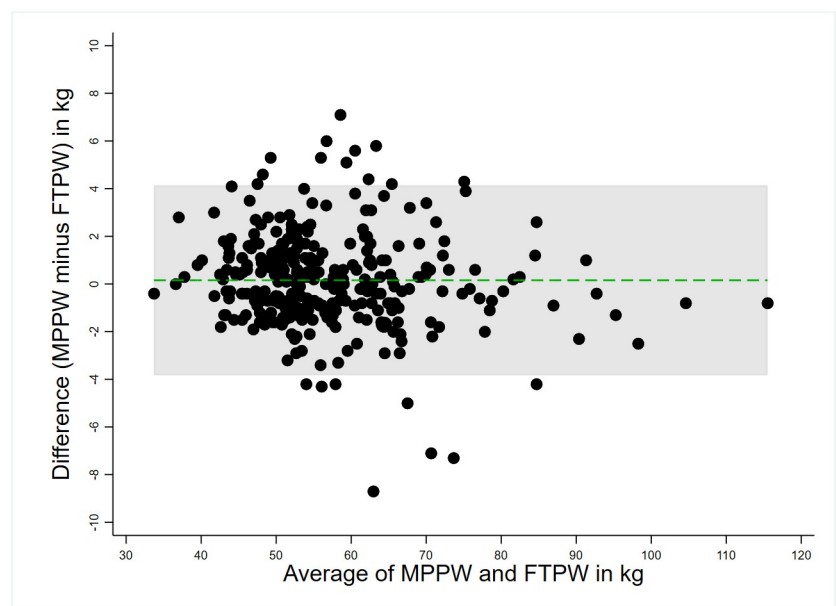

**Fig 3. Bland & Altman plot comparing the first trimester weight (FTPW) to the measured pre-pregnancy weight (MPPW).** RECIPAL cohort, Southern Benin, 2014–2017 (n = 302).

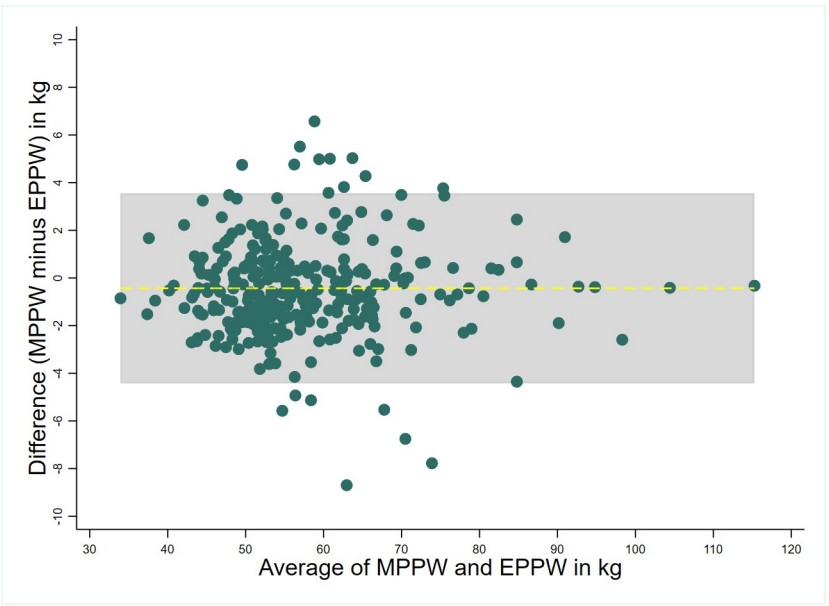

**Fig 4. Bland & Altman plot comparing the estimated pre-pregnancy weight using Thomas formula (EPPW) to the measured pre-pregnancy weight (MPPW).** RECIPAL cohort, Southern Benin, 2014–2017 (n = 302).

marginally underestimated by the FTPW, with a mean difference of -0.16 kg. Conversely, the EPPW exhibits a slight overestimation of the MPPW, with the mean difference calculated at +0.43 kg. Globally, the FTPW and EPPW showed a good agreement and concordance with the MPPW (Table 2). Nevertheless, there exists a notable degree of variation with the individual differences (MPPW minus proxies) ranging from -8.70 to +7.10 kg, as well as wide limits of agreement, as delineated in Table 2. According to the Kappa analyses, the agreement between the distribution in BMI categories using the MPPW and the ones using the two proxies was high (S2 Table), the kappa coefficients being greater than 0.80 for both (Table 2).

Table 3, as well as S3 and S4 Tables, demonstrate subtle fluctuations in the results of the proxies against the reference weight with respect to gestational age across the first trimester of

**Table 2. Comparisons between the MPPW[a] and the two proxies, RECIPAL cohort, Southern Benin, 2014–2017, n = 302.**

| Parameters | | Comparison between methods | |
|---|---|---|---|
| | | FTPW[b] vs. MPPW[a] | EPPW[c] vs. MPPW[a] |
| **Bland and Altman method parameters** | Range of differences (kg) | -8.70; +7.10 | -8.69; +6.68 |
| | Mean difference (95% CI), in kg | +0.16 (-0.08; +0.39) | -0.43(-0.66; -0.20) |
| | Limits of agreement (kg) | -3.83; +4.14 | -4.42; +3.55 |
| | 95% CI of lower limit of agreement (kg) | -4.25; -3.46 | -4.84; -4.05 |
| | 95% CI of upper limit of agreement (kg) | +3.77; +4.56 | +3.18; +3.97 |
| | Proportion of weights outside the limits of agreement | 6.62% | 5.96% |
| **Lin Concordance Correlation Coefficient between weights assessments (IC 95%)** | | 0.98(0.98, 0.99) | 0.98(0.98, 0.99) |
| **Concordance of BMI categorization Kappa Coefficient (P-value)** | | 0.84(P<0.0001) | 0.86(P<0.0001) |

[a]: Measured Pre-pregnancy Weight,

[b]: First Trimester Pregnancy Weight,

[c]: Estimated Pre-pregnancy Weight using Thomas et al formula.

**Table 3. Differences between the MPPW[b] and the two proxies by categories of gestational age and time of measurement of MPPW[b], RECIPAL cohort, Southern Benin, 2014–2017.**

| Parameters | n | Weight (kg) by method of estimation | | | | | | Differences between methods | | | |
|---|---|---|---|---|---|---|---|---|---|---|---|
| | | MPPW[b] | | FTPW[c] | | EPPW[d] | | FTPW[c] vs MPPW[b] | | EPPW[d] vs MPPW[b] | |
| | | Mean | SD | Mean | SD | Mean | SD | Crude Diff | 95%CI | Crude Diff | 95%CI |
| | | | | | | | | P<0.0001 | | | |
| All | 2226[1] | 57.7 | 11.3 | 57.6 | 11.5 | 58.0 | 11.3 | -0.11 | -0.33, +0.12 | +0.30 | +0.07, +0.53 |
| [a] GA Categories (weeks) for FTPW assessment | 2226[1] | | | | | | | P<0.0001 | | | |
| GA ≤ 5 | 354 | 57.9 | 11.8 | 57.9 | 12.1 | 58.6 | 11.8 | -0.07 | -0.44, +0.30 | +0.66 | +0.29, +1.04 |
| 5 < GA <7 | 288 | 59.4 | 11.8 | 59.3 | 12.2 | 59.9 | 11.9 | -0.13 | -0.56, +0.31 | +0.43 | -0.00, +0.86 |
| 7 ≤ GA< 9 | 408 | 57.2 | 10.7 | 57.1 | 11.0 | 57.6 | 10.8 | -0.13 | -0.50, +0.24 | +0.38 | +0.01, + 0.76 |
| 9 ≤ GA < 11 | 363 | 57.5 | 10.8 | 57.2 | 10.6 | 57.6 | 10.4 | -0.37 | -0.81, +0.07 | +0.06 | -0.39, + 0.50 |
| 11 ≤ GA < 13 | 468 | 57.4 | 11.1 | 57.2 | 11.6 | 57.5 | 11.4 | -0.21 | -0.64, +0.22 | +0.03 | -0.40, +0.46 |
| 13 ≤ GA < 14 | 345 | 57.3 | 12.1 | 57.6 | 11.9 | 57.7 | 11.6 | +0.30 | -0.17, +0.78 | +0.34 | -0.13, +0.81 |
| MPPW measurement time before pregnancy | 742[2] | | | | | | | P<0.0001 | | | |
| < 3 months | 591 | 57.0 | 11.2 | 56.8 | 11.4 | 57.2 | 11.1 | -0.16 | -0.41, +0.09 | +0.26 | +0.01, +0.51 |
| ≥ 3 months | 151 | 60.7 | 11.2 | 60.8 | 11.6 | 61.2 | 11.4 | +0.10 | -0.43, +0.64 | +0.45 | -0.10, +0.99 |
| | | | | | | | | | | | |

[a]: Gestational Age,

[b]: Measured Pre-pregnancy Weight,

[c]: First Trimester Pregnancy Weight,

[d]: Estimated Pre-pregnancy Weight using Thomas et al formula.

[1] Unit of analysis: Woman x method x visit.

[2] Unit of analysis: Women x method.

pregnancy. Table 3 shows that the differences between MPPW and each of its proxies (FTPW and EPPW) are statistically significant and not influenced by gestational age during the first trimester of pregnancy. Occasionally, a slightly better agreement is observed for EPPW, while in other instances, FTPW yields slightly better results. However, these fluctuations remain generally marginal. Regardless of when weight is measured in the first trimester of pregnancy, Thomas formula slightly overestimates pre-pregnancy weight, and first-trimester weight measured before 13±1 weeks offers a minor underestimate of pre-pregnancy weight (S3 Table). S4 Table outlines the Bland and Altman comparison parameters according to gestational age categories and results suggest minimal sensitivity to gestational age of the parameters of agreement. When the analysis also included the women whose MPPW was measured more than three months before pregnancy, the precision of the approximation method (FTPW or EPPW) was reduced without impacting the direction of variation (under- or over-estimation), regardless of the method used (Table 3 and S5 Table).

## Discussion

In this study, we assessed the performance of two proxies of pre-pregnancy weight, using data of a pre-pregnancy cohort from southern Benin, West-Africa. Our results showed that a weight measured no later than 13±1 weeks of gestation can serve as a reliable proxy for pre-pregnancy weight. The first trimester of pregnancy weight and the estimated pre-pregnancy weight using Thomas *et al.* formula gave close estimates and similar values of the parameters of Bland & Altman agreement and of concordance correlation coefficients, with slight average underestimation of -0.16 kg for FTPW and overestimation of 0,43 kg for EPPW. The

gestational age at which the weight was measured during the first trimester of pregnancy did not significantly change the agreement nor the concordance correlation of each proxy with the measured pre-pregnancy weight (MPPW). Overall, there was no advantage in using Thomas *et al* formula to estimate the pre-pregnancy weight instead of using directly a weight measured in the first trimester of pregnancy in our study.

As for the median spline, we observed a flat slope of the average weight gain in the first trimester of pregnancy, with approximately half of the points on each side of the regression line (Fig 2). This aligns with the observed occurrence of a minor underestimation when using first-trimester weight as a proxy for pre-pregnancy weight in our study population. This tendency to weight loss in the first trimester for approximately half of the women could be attributed to sympathetic disturbances, often followed by symptoms such as vomiting and loss of appetite, as reported in several studies, including a meta-analysis [30,31].

In the scientific literature we identified three papers studying pre-pregnancy weight proxies in different settings, thus allowing some comparisons with the results of our study in a sub-Saharan African context. In the validation study of the model developed by Thomas et al., data from 51 American women from the Fit for Delivery (FFD) cohort who had a pre-pregnancy weight measured within a 6-month interval before pregnancy was used [16]. The authors obtained an average underestimation of 0.68 kg, while we observed an average overestimation of 0.43 kg. These disparities between the two studies can be explained by the combined influence of several factors. First, our study boasts a sample size nearly six times larger than that of Thomas et al. (302 vs. 51), coupled with weight measurements obtained within three months before pregnancy in our study versus six months in Thomas *et al.* study. Second, gestational age was objectively determined for each participant in our study using ultrasound scan for pregnancy dating in first trimester, whereas it was self-reported by participants in the other study. Last, the women in our study were assessed much earlier in pregnancy, with an average gestational age of 7±2.4 weeks, in comparison to 13.5±1.8 weeks in the study conducted by Thomas et al. Moreover, the two study populations originate from distinct contexts—developing countries versus developed countries—potentially imbuing them with intrinsic characteristics unique to each respective setting. The second study is the one conducted by Inskip et al. on a cohort of 198 women from Southampton, UK [17]. Our results are similar to those of Inskip et al., who also observed a tendency of Thomas' formula to overestimate, on average, the measured pre-pregnancy weight (by a mean of 0.88 kg, compared to 0.43 kg in our study) [17]. However, regarding the use of the weight measured during the first trimester of pregnancy to approximate the pre-pregnancy weight, we observed an underestimation by an average of -0.16 kg, whereas Inskip et al. found an overestimation by an average of 0.88 kg. This difference between Inskip's results and ours could be attributable to a more frequent vomiting and weight loss during the first trimester of pregnancy in our sample, as commented above. The third study was carried out on a Chinese pre-pregnancy cohort of 474 women and focussed only on how well weight measured in the first trimester of pregnancy approximate the pre-pregnancy weight [14]. The results showed an overestimation of the actual pre-pregnancy weight of 1.3 kg on average, but the quality of the agreement and the concordance correlation coefficients were lower than ours, as well as than those obtained by the two other studies (by Thomas et al. and Inskip et al.). This was likely due to the fact that pre-pregnancy weight measurement in the Chinese study occurred at a median of 17.1 weeks before pregnancy (interquartile range 5.3–46.9 weeks), thus highlighting that to be considered as a reference it is preferable that a pre-pregnancy weight is measured shortly enough before the pregnancy starts. Interestingly enough, however, this Chinese study also examined whether the magnitude of the bias in the estimation of the pre-pregnancy weight was influenced by the timing of weight measurement during the first trimester of pregnancy and, as we did, the authors

showed that the bias remained of similar size and the concordance correlations coefficients remained of similar values, whatever the gestational age during the first trimester.

The practical implications of our results are linked to the fact that an accurate determination of preconception weight is crucial for effective pregnancy monitoring, serving two primary purposes: firstly, it allows a more precise quantification of total gestational weight gain, provided that the woman's weight at the end of pregnancy is available of course. Secondly, pre-pregnancy weight is required for classifying women into the appropriate pre-pregnancy BMI category, enabling the evaluation of weight gain adequacy in accordance with established IOM recommendations [3]. Our analysis suggests that the first trimester weight can serve as a viable proxy for pre-pregnancy weight at a population level and for research endeavours, given the relatively small average differences observed in the study and also the good concordance in BMI categories confirmed by the kappa analysis. It is important to note, however, that disparities between the proxies and the reference weight at the individual level exhibited a range from -8.7kg to +7.1kg in our study. Consequently, it is likely that employing this proxy may result in a subset of women being erroneously categorized into inappropriate BMI classifications, thereby potentially resulting in misinterpretations of weight gain and subsequent decision-making errors at the individual level [32]. Therefore, regular individual-level monitoring is essential to mitigate the risk of such errors by ensuring the availability of adequate information for individual decision-making purposes.

Our study has several important strengths. The original design of the prospective cohort study, women benefited from an accurate pre-pregnancy weight measurement, within three months before pregnancy for the majority of them, then from an accurate dating of the pregnancy by early ultrasound scan and finally from an early weight measurement during the first trimester of pregnancy. However, there are some limitations to consider. As the study was conducted on a specific population in the southern region of Benin, it is important to acknowledge that the results cannot be generalized to the entire Beninese population. As we present in this paper the results of a secondary analysis that was not foreseen when the project was drafted, we missed some information which would have been interesting to evaluate in our context, such as data on self-reported pre-pregnancy weight. In addition, there were some differences between women retained for the main analysis and those who were excluded (S1 Table). These differences, however, were clearly due to the fact more women from the district of So-Ava than from the district of Akassato were excluded. This was because the study started earlier in the So-Ava district, so the follow-up was longer, making more women having had a pre-pregnancy weight measured more than 3 months before pregnancy and being, therefore, excluded. But there is no obvious reason to think that this could introduce a bias in our analysis. While caution should be exercised in generalizing the findings, the meticulousness of the study design and methodology enhances the reliability and validity of the results within the studied population.

There are however many technical questions still pending about the assessment of weight gain during pregnancy [20] and further studies in different contexts need to be carried out before the IOM recommendations can be revised/adapted. In this respect it is highly welcomed that WHO recently launched a multi-country database project to develop global gestational weight gain standards [33]. This will be foremost in addressing the notable gap in evidence-based public health tools for gestational weight gain monitoring and to issue generalizable recommendations.

## Supporting information

**S1 Table. Sociodemographic and anthropometric characteristics of the 302 selected women compared to the 109 excluded ones in the analysis; RECIPAL study in Benin,**

2014–2017.
(DOCX)

**S2 Table. Changes in BMI categories according to the pre-pregnancy weight used to calculate the BMI of each study participants (n = 302); RECIPAL study, Benin, 2014–2017.**
(DOCX)

**S3 Table. Description of the First Trimester of Pregnancy Weight (FTPW) and the corresponding Estimated Pre-pregnancy Weight using the proposed formula of Thomas et al. (EPPW), according to the gestational age at measurement, RECIPAL study, Benin, 2014–2017.**
(DOCX)

**S4 Table. Gross value (kg) and percentage variation in the First Trimester of Pregnancy Weight (FTPW) and the corresponding Estimated Pre-pregnancy Weight using the proposed formula of Thomas et al. (EPPW, RECIPAL study, Benin, 2014–2017.**
(DOCX)

**S5 Table. Bland and Altman comparison parameters of the MPPW to the FTPW and to the corresponding EPPW, by gestational age ranges in the first trimester of pregnancy and according to the timing of measurement of the MPPW; RECIPAL study, Benin, 2014–2017.**
(DOCX)

**S1 File.**
(DOC)

## Acknowledgments

The authors thank all the local communities of Sô-Ava and Akassato in Benin who took part in this study; all RECIPAL project technical team, the midwives, nurses, community-health workers for the hard work of recruiting and following the study participants.

## Author Contributions

**Conceptualization:** Emmanuel Yovo, Manfred Accrombessi, Valérie Briand, Jules Alao, Jennifer Zeitlin, Pierre Traissac, Yves Martin-Prevel.

**Data curation:** Emmanuel Yovo, Manfred Accrombessi, Gino Agbota, Cornelia Hounkonnou.

**Formal analysis:** Emmanuel Yovo, Pierre Traissac.

**Funding acquisition:** Valérie Briand, Yves Martin-Prevel.

**Investigation:** Emmanuel Yovo, Manfred Accrombessi, Gino Agbota, Cornelia Hounkonnou, Yves Martin-Prevel.

**Methodology:** Emmanuel Yovo, Gino Agbota, Jules Alao, Jennifer Zeitlin, Pierre Traissac, Yves Martin-Prevel.

**Project administration:** Manfred Accrombessi, Valérie Briand, Yves Martin-Prevel.

**Resources:** Yves Martin-Prevel.

**Software:** Emmanuel Yovo, Jennifer Zeitlin, Pierre Traissac.

**Supervision:** Emmanuel Yovo, Manfred Accrombessi, Valérie Briand, Gino Agbota, Jules Alao, Jennifer Zeitlin, Pierre Traissac, Yves Martin-Prevel.

**Validation:** Manfred Accrombessi, Valérie Briand, Jules Alao, Jennifer Zeitlin, Pierre Traissac, Yves Martin-Prevel.

**Visualization:** Emmanuel Yovo, Cornelia Hounkonnou, Jennifer Zeitlin, Pierre Traissac, Yves Martin-Prevel.

**Writing – original draft:** Emmanuel Yovo.

**Writing – review & editing:** Emmanuel Yovo, Manfred Accrombessi, Valérie Briand, Gino Agbota, Cornelia Hounkonnou, Jules Alao, Jennifer Zeitlin, Pierre Traissac, Yves Martin-Prevel.

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
