## [Decision Letter · Decision Letter 0]

8 Aug 2024

PONE-D-24-21305Comparison of two proxies for the preconception weight using data from a pre-pregnancy cohort in Benin: weight measured in the first trimester of pregnancy vs estimated by Thomas’ formula.PLOS ONE

Dear Dr. YOVO,

Thank you for submitting your manuscript to PLOS ONE. After careful consideration, we feel that it has merit but does not fully meet PLOS ONE’s publication criteria as it currently stands. Therefore, we invite you to submit a revised version of the manuscript that addresses the points raised during the review process.

 Responses to my comments and those of the reviewers is required.

We look forward to receiving your revised manuscript.

Kind regards,

Emma K. Kalk

Academic Editor

PLOS ONE

Journal Requirements:

3. Thank you for stating the following financial disclosure: "This work was supported by the French Agence Nationale de la Recherche (grant number ANR-13-JSV1-0004) and the Foundation Simone Beer under the auspices of the Fondation de France (grant number 00074147). EY received ARTS (Allocations de Recherche pour une Thèse au Sud), the original French name of a PhD grants programme from the French National Research Institute for Sustainable Development (IRD) for a PhD study at Montpellier University".

Additional Editor Comments:

The authors present an evaluation of two methods to determine pre-pregnancy weight, first trimester pregnancy weight and the use of Thomas’ formula, comparing agreement between the methods and between each method and true pre-pregnancy weight in a cohort of women from Benin. First trimester weights slightly underestimated pre-pregnancy weight (by mean ~150g) while the application of Thomas’ formula overestimated pre-pregnancy weight by a small bit greater degree (mean 430g). ~6% of weights were outside of the limits of the agreement between the method and the measured pre-pregnancy weight. Having a measured pre-pregnancy weight is a strength as is applying the analyses in sub-Saharan Africa.

The manuscript is well-written and clear. The variables are clearly defined.

1. As noted by reviewer 2, please could you clarify how gestational weight gain was determined?

2. Some of the language could be clarified. E.g., ‘concubinage’ is not a conventional term. Do you mean unmarried?

3. The limitations of the study could be discussed in more detail.

Row 195: wg – does this indicate week?

199. 201: language is not scientific. You have the numbers so no need for, ‘around’. In addition, use of number (2) and words (three). How numbers are expressed should be consistent and aligned with PLoSone guidelines.

Table 1 – will need to be formatted in line with PLoS guidelines.

Parity. What is Unit?

Is measure pre-pregnancy weight at 3 months pre-pregnancy?

Reviewers' comments:

Reviewer's Responses to Questions

**Comments to the Author**

1. Is the manuscript technically sound, and do the data support the conclusions?

Reviewer #1: Yes

Reviewer #2: No

2. Has the statistical analysis been performed appropriately and rigorously? 

Reviewer #1: Yes

Reviewer #2: No

3. Have the authors made all data underlying the findings in their manuscript fully available?

Reviewer #1: Yes

Reviewer #2: No

4. Is the manuscript presented in an intelligible fashion and written in standard English?

Reviewer #1: Yes

Reviewer #2: Yes

5. Review Comments to the Author

Reviewer #1: This is a well-written and interesting study. The authors provide convincing rationale for why to examine this question in a lower income, sub-Saharan country (this question has been examined in higher-income countries). The study upon which these secondary analyses were based was well-designed, with frequent measured pre-pregnancy weights.

1) Line 110: I believe the authors mean "its" rather than "it's"

2) The authors appear to have missed one previous study that has investigated this question, and it would be important to compare and contrast the current findings with this previous study.

Krukowski, R. A., West, D. S., DiCarlo, M., Shankar, K., Cleves, M. A., Saylors, M. E., & Andres, A. (2016). Are early first trimester weights valid proxies for preconception weight?. BMC pregnancy and childbirth, 16, 1-6.

Reviewer #2: I appreciate the opportunity to review this paper. The problem of the mother’s weight at the beginning of pregnancy is important, especially in weight-gain studies, and has been understudied.

This study has one major positive point which is a measured pre-pregnancy weight (within 3 months prior to conception) to be used as a gold standard. However, two major points are worth attention.

1. The authors used gestational weight gain (GWG) to identify the gestational age at which weight gain starts to become significant, but they did not explain how GWG was calculated. This poses a substantial problem because the point will change if you use one value (pre-pregnancy weight) or the other (first-trimester weight) to calculate GWG.

2. The major issue is that it is not clear why not deriving an equation in the study to estimate the weight at conception based on first-trimester weight. Applying an equation developed for a very different North American population (and in a low sample) in this study is not well justified, especially because the authors had a ‘gold standard’ (the weight measured before the conception). Why not derive the ‘error’ from the first-trimester weight from the measured pre-pregnancy weight (the ‘gold standard’) in those individuals who had both and use this equation to predict the pre-pregnancy weight for those with first-trimester weight only? Considering that the authors concluded that the equation does not represent any gain for this population (it might not apply to them - as expected!), it is unclear to me the rationale for doing this analysis in the first place.  

In addition, my other comments are:

3. Pre-pregnancy weight is usually relevant for the calculation of pre-pregnancy BMI. Several studies in the field have looked at how using different weights (self-reported, measured pre-pregnancy, first trimester, etc) would impact pre-pregnancy BMI classification. This is usually done by calculating the Kappa coefficient. This is a relevant analysis that needs to be included.

4. Do the authors have collected self-reported pre-pregnancy weight? If so, why not evaluate the agreement of that with the gold standard as well?

5. It is not correct to use Pearson’s correlation coefficient to evaluate agreement between two variables. Please refer to 10.1016/j.theriogenology.2010.01.003 for more information.

6. How do the 302 women selected for this study compare to the 411 women who became pregnant in the RECIPAL cohort?

7. The end of the statistical analysis section is not clear.

6. PLOS authors have the option to publish the peer review history of their article (what does this mean?). If published, this will include your full peer review and any attached files.

Reviewer #1: No

Reviewer #2: No

---

## [Author Response · Author response to Decision Letter 0]

10 Sep 2024

Dear editor,

Thank you for considering our manuscript PONE-D-24-21305, entitled “Comparison of two proxies for the preconception weight using data from a pre-pregnancy cohort in Benin: weight measured in the first trimester of pregnancy vs estimated by Thomas’ formula”. 

We have revised the issues brought up by the reviewers and a point-by-point response can be found below. Please note that, as part of the process of taking comments into account, we have added two tables to the supporting information files. 

We agree with the amended statement about the role of the funders that you suggested: "The funders had no role in study design, data collection and analysis, decision to publish, or preparation of the manuscript. But the IRD who funded the PhD grant."

Finally, as requested, we are submitting a clean and a track changes versions of the manuscript. Please note that in the present rebuttal letter all line numbers indicating where changes have been made in the manuscript refer to the version with track changes. 

With kind regards,

Dr Emmanuel Koffi YOVO

Clinical Research Institute of Benin (IRCB)

04 BP 1114, Abomey-Calavi

Email: emkoffiyovo@gmail.com

Tel: +229 97 42 08 01

Response to the academic editor

Journal Requirements:

The format of chapter headings, tables and file names has been revised to conform to the journal's requirements.

The formatting of authors affiliations has been checked and modified when necessary to comply with the journal’s requirements. 

The questionnaire has been completed and submitted as supporting information as requested.

3. Thank you for stating the following financial disclosure: "This work was supported by the French Agence Nationale de la Recherche (grant number ANR-13-JSV1-0004) and the Foundation Simone Beer under the auspices of the Fondation de France (grant number 00074147). EY received ARTS (Allocations de Recherche pour une Thèse au Sud), the original French name of a PhD grants programme from the French National Research Institute for Sustainable Development (IRD) for doing his PhD at Montpellier University".

Please state what role the funders took in the study. If the funders had no role, please state: "The funders had no role in study design, data collection and analysis, decision to publish, or preparation of the manuscript. But the IRD who funded the PhD grant "

We agree with the above suggested statement, and we have included it in our cover letter as requested.

Thanks, the phrase has been reformulated for more clarity, by referring to Figure 2 which contains all the necessary data (lines 383 to 387, in the revised version with track changes)

Additional Editor Comments:

 The authors present an evaluation of two methods to determine pre-pregnancy weight, first trimester pregnancy weight and the use of Thomas’ formula, comparing agreement between the methods and between each method and true pre-pregnancy weight in a cohort of women from Benin. First trimester weights slightly underestimated pre-pregnancy weight (by mean ~150g) while the application of Thomas’ formula overestimated pre-pregnancy weight by a small bit greater degree (mean 430g). ~6% of weights were outside of the limits of the agreement between the method and the measured pre-pregnancy weight. Having a measured pre-pregnancy weight is a strength as is applying the analyses in sub-Saharan Africa.

The manuscript is well-written and clear. The variables are clearly defined.

1. As noted by reviewer 2, please could you clarify how gestational weight gain was determined?

Thanks for the request. Please see below my responses to reviewer 2 comment regarding this point.

2. Some of the language could be clarified. E.g., ‘concubinage’ is not a conventional term. Do you mean unmarried?

Thanks for the suggestion. Concubinage has been replaced by "Unmarried cohabitation” in Table1 (page 14 in the manuscript version with track changes).

3. The limitations of the study could be discussed in more detail.

The limitations are now more discussed, particularly after considering the reviewer 2 comments. More specifically, two points were added: Firstly, about the unavailability of data on self-reported pre-pregnancy weight, we specified that this was because we present here a secondary analysis of the RECIPAL study, which didn’t plan to analyze such kind of data. Secondly, about the representativeness of the women considered in the analyses, versus those who were excluded, we now present the results of the comparison between these groups in a new table (S1 Table). Please see also our reply to comment #6 of reviewer #2 below and lines 460 to 472.

 Row 195: wg – does this indicate week?

Yes, wg stands for weeks of gestation. Please note that this abbreviation was previously defined (lines 155-156).

199. 201: language is not scientific. You have the numbers so no need for, ‘around’. In addition, use of number (2) and words (three). How numbers are expressed should be consistent and aligned with PLoSone guidelines.

The word “around” has been deleted (line 285 and 288) and numbers expression has been corrected to align with the journal’s guidelines (lines 285 and 289).

Table 1 – will need to be formatted in line with PLoS guidelines.

Table1 formatting has been corrected to align with the journal’s guidelines.

Parity. What is Unit?

Parity is the number of deliveries the mother got after 20 weeks of pregnancy before her inclusion in the study.

Is measure pre-pregnancy weight at 3 months pre-pregnancy?

Not exactly; indeed, the pre-pregnancy weight was measured within 3 months’ time before pregnancy start. We understand that this comment is linked to the lack of clarity at the end of the statistical paragraph. All this paragraph has now been reworded as requested by reviewer 2 (lines 175 to 216).

Response to Reviewer # 1

This is a well-written and interesting study. The authors provide convincing rationale for why to examine this question in a lower income, sub-Saharan country (this question has been examined in higher-income countries). The study upon which these secondary analyses were based was well-designed, with frequent measured pre-pregnancy weights.

1) Line 110: I believe the authors mean "its" rather than "it's"

Thank you very much for the kind words and for accepting to review our paper. This typo has been corrected (line 134, in the version with track changes of the manuscript).

2) The authors appear to have missed one previous study that has investigated this question, and it would be important to compare and contrast the current findings with this previous study.

Krukowski, R. A., West, D. S., DiCarlo, M., Shankar, K., Cleves, M. A., Saylors, M. E., & Andres, A. (2016). Are early first trimester weights valid proxies for preconception weight? BMC pregnancy and childbirth, 16, 1-6.

Thanks for bringing this article to our attention. 

Response to Reviewer # 2

I appreciate the opportunity to review this paper. The problem of the mother’s weight at the beginning of pregnancy is important, especially in weight-gain studies, and has been understudied.

This study has one major positive point which is a measured pre-pregnancy weight (within 3 months prior to conception) to be used as a gold standard. However, two major points are worth attention.

1. The authors used gestational weight gain (GWG) to identify the gestational age at which weight gain starts to become significant, but they did not explain how GWG was calculated. This poses a substantial problem because the point will change if you use one value (pre-pregnancy weight) or the other (first-trimester weight) to calculate GWG.

Thank you for your positive comment. The gestational weight gains were calculated by subtracting the measured pre-pregnancy weight from each weight measurement during pregnancy. This is now clarified (lines 158-160, in the version with track changes of the manuscript).

2. The major issue is that it is not clear why not deriving an equation in the study to estimate the weight at conception based on first-trimester weight. Applying an equation developed for a very different North American population (and in a low sample) in this study is not well justified, especially because the authors had a ‘gold standard’ (the weight measured before the conception). Why not derive the ‘error’ from the first-trimester weight from the measured pre-pregnancy weight (the ‘gold standard’) in those individuals who had both and use this equation to predict the pre-pregnancy weight for those with first-trimester weight only? Considering that the authors concluded that the equation does not represent any gain for this population (it might not apply to them - as expected!), it is unclear to me the rationale for doing this analysis in the first place. 

Thanks for the suggestion. Methodologically, your suggestion is interesting but from a different perspective. This approach would indeed allow us to have an equation specifically adapted to the context of our study. However, the chosen option for the analyses in this study was primarily to assess the performance of two proxies, including Thomas’ equation, in our population, then to compare these performances with those obtained by similar studies conducted in the context of developed countries. Therefore, we limited ourselves to examining whether, conceptually, this prediction of pre-pregnancy weight based on the characteristics of the woman considered in Thomas’ equation could pave the way for adopting such an approach, like what is done with the IOM references for gestational weight gain, for example.

3. Pre-pregnancy weight is usually relevant for the calculation of pre-pregnancy BMI. Several studies in the field have looked at how using different weights (self-reported, measured pre-pregnancy, first trimester, etc.) would impact pre-pregnancy BMI classification. This is usually done by calculating the Kappa coefficient. This is a relevant analysis that needs to be included

Thanks for the useful suggestion. Kappa coefficients analysis is now included in the method section (line 193 -196), in the results (lines 325-328 and Table 2 at page 16) and in the discussion (lines 444-445). 

4. Do the authors have collected self-reported pre-pregnancy weight? If so, why not evaluate the agreement of that with the gold standard as well?

Unfortunately, self-reported pre-pregnancy weight was not collected. Indeed, the secondary analysis that we present in our manuscript was not foreseen when the study was set up. We have included this point in the limitations section (lines 460-462).

5. It is not correct to use Pearson’s correlation coefficient to evaluate agreement between two variables. Please refer to 10.1016/j.theriogenology.2010.01.003 for more information.

Thanks for sharing that interesting and useful paper. The Pearson’s correlation coefficient components have been removed from the methods (line 235) and the results (line 314 and Table 2 at page 16). Consequently, we also suppressed any reference to Pearson’s correlation coefficient when we compared our results to those from other studies (lines 397 to 400 and lines 415-417). Please note that for more clarity we have also added the word “concordance” before “correlation coefficients” since we still use the Lin correlation coefficients for gauging the agreement (e.g Table 2 at page 16 and lines 374, 377, 426, 434).

6. How do the 302 women selected for this study compare to the 411 women who became pregnant in the RECIPAL cohort?

Thanks for the comment. A table comparing the 302 selected women to the 109 unselected women of RECIPAL study was added as a supplementary table (please see S1 Table in the supporting information file and lines 297-299 in the results section). The women who were not included in the analysis tended to be taller and larger in body size. They were more often from the first site of the study (So-Ava) than from the second one (Akassato) and were more frequently of the Toffin ethnic group. The RECIPAL study was initially planned to be conducted exclusively within the lake-dwelling community of So-Ava. A sample of 500 pregnant women was targeted but it happened to be difficult to reach this goal within the sole So-Ava community for several reasons, including cases of refusal, among others. It was therefore decided to extend the study area to a neighboring locality (Akassato). The women from So-Ava, who are primarily of the Toffin ethnicity, were therefore followed for a longer period during the pre-pregnancy period (since the study began with them), and for some of them this led their latest pre-pregnancy weight measurement to be more than three months old when they became pregnant because weights were no longer measured after one year of pre-pregnancy follow-up. There were also more cases of withdrawal of consent among the Toffin women, who also happened to be larger in body size. These differences were therefore only due to a different study timing between the two locations and there is no obvious reason to think that this could have introduced a bias in our analysis. A shortened version of the above explanation has been added in the limitations section of the discussion (lines 462 to 469).

7. The end of the statistical analysis section is not clear.

We have entirely reworded the statistical analysis section with a special focus on the last paragraph to make it clearer, as suggested by the reviewer. This gave us the opportunity to enhance readability throughout the statistical analyses section (lines 175 to 216).

---

## [Editor Report · Decision Letter 1]

1 Oct 2024

PONE-D-24-21305R1Comparison of two proxies for the preconception weight using data from a pre-pregnancy cohort in Benin: weight measured in the first trimester of pregnancy vs estimated by Thomas’ formula.PLOS ONE

Dear Dr. Yovo,

Thank you for submitting your manuscript to PLOS ONE. After careful consideration, we feel that it has merit but does not fully meet PLOS ONE’s publication criteria as it currently stands. Therefore, we invite you to submit a revised version of the manuscript that addresses the points raised during the review process.

We look forward to receiving your revised manuscript.

Kind regards,

Emma K. Kalk

Academic Editor

PLOS ONE

Journal Requirements:

Additional Editor Comments:

The reviewer’s comments have been adequately addressed. Minor comments remain with respect to presentation.

Delete ‘rather’ wherever it occurs in the text. It is unscientific.

There is no need to include the STATA code.

198: delete *sort of*

200 delete *thus*

212: delete *as above*

216: delete *also*

290: delete the *“”*

297: don’t capitalize *Sociodemographic*

453: delete *In particular, thanks to*

462: delete *for example*

---

## [Author Response · Author response to Decision Letter 1]

11 Oct 2024

Dear editor,

We sincerely appreciate the opportunity to resubmit our revised manuscript, entitled “Comparison of two proxies for the preconception weight using data from a pre-pregnancy cohort in Benin: weight measured in the first trimester of pregnancy vs estimated by Thomas’ formula”. 

We have carefully considered all the journal requirements and your additional comments to further strengthen the quality of our work. 

Below, we provide detailed responses to each of the suggestions and explain the specific changes made to the manuscript. 

We trust that these revisions address the remaining concerns and enhance the clarity and robustness of our study. 

With kind regards,

Dr Emmanuel Koffi YOVO

Clinical Research Institute of Benin (IRCB)

04 BP 1114, Abomey-Calavi

Email: emkoffiyovo@gmail.com

Tel: +229 97 42 08 01

Response to the academic editor

Journal Requirements:

Please review your reference list to ensure that it is complete and correct. If you have cited papers that have been retracted, please include the rationale for doing so in the manuscript text or remove these references and replace them with relevant current references. Any changes to the reference list should be mentioned in the rebuttal letter that accompanies your revised manuscript. If you need to cite a retracted article, indicate the article’s retracted status in the References list and also include a citation and full reference for the retraction notice.

We reviewed all references, the context, and the rationale for their inclusion in the article, starting from the original submitted version to the first revised version. The clarifications are as follows:

1. In the initial version of the submitted manuscript, reference number 20 was automatically re-identified by Zotero reference management software as reference 33. During the first revision process, this error was corrected, and reference number 20 was uniquely assigned to this article, which has been consistently maintained in the paper.

2. Additionally, during the revision process, reference 27, which was cited in the Methods section of the initial submitted version, provided details on the restricted cubic splines method used for the analysis. Given that this method is already well-known, the reference was ultimately deemed unnecessary and was removed from the revised version. 

3. In accordance with the reviewer 2 suggestion regarding the Kappa coefficient, we have added reference 29 to justify the inclusion of Kappa coefficient analyses in the revised version of the manuscript.

We have made a concerted effort to thoroughly address the reviewers' comments, but we inadvertently omitted to clarify these changes regarding references, which were made on our own initiative. Thank you for giving us the opportunity to clarify this.

Additional Editor Comments:

1) The reviewer’s comments have been adequately addressed. Minor comments remain with respect to presentation.

Thanks.

We have taken all the suggestions into account, and the lines indicated below allow for verification in the tracked changes version of the manuscript.

2) Delete ‘rather’ wherever it occurs in the text. It is unscientific.

“Rather” deleted on lines: 45; 80; 221; 290 and 316. 

3) There is no need to include the STATA code.

STATA codes are removed from the manuscript on lines: 158; 164; 167; 175-176 and 180. 

All the following suggestions have been considered: 

198: delete sort of (Done, line 170)

200 delete thus (Done, line 172)

212: delete as above (Done, line 184)

216: delete also (Done, line 188)

290: delete the “” (Done, line 207)

297: don’t capitalize Sociodemographic (Done, line 212)

453: delete In particular, thanks to (Done, line 355)

462: delete for example (Done, line 364)

---

## [Editor Report · Decision Letter 2]

15 Oct 2024

Comparison of two proxies for the preconception weight using data from a pre-pregnancy cohort in Benin: weight measured in the first trimester of pregnancy vs estimated by Thomas’ formula.

PONE-D-24-21305R2

Dear Dr. Yovo,

We’re pleased to inform you that your manuscript has been judged scientifically suitable for publication and will be formally accepted for publication once it meets all outstanding technical requirements.

Kind regards,

Emma K. Kalk

Academic Editor

PLOS ONE
---

## [Editor Report · Acceptance letter]

23 Oct 2024

PONE-D-24-21305R2 

PLOS ONE

Dear Dr. Yovo, 

I'm pleased to inform you that your manuscript has been deemed suitable for publication in PLOS ONE. Congratulations! Your manuscript is now being handed over to our production team.

Kind regards, 

on behalf of

Dr. Emma K. Kalk 

Academic Editor

PLOS ONE